# 3D RBSM Analysis of Bond Degradation in Corroded Reinforced Concrete as Observed Using Digital Image Correlation

**DOI:** 10.3390/ma15186470

**Published:** 2022-09-18

**Authors:** Kumar Avadh, Kohei Nagai

**Affiliations:** Institute of Industrial Science, The University of Tokyo, Tokyo 1538505, Japan

**Keywords:** corrosion, de-bonding, digital image correlation, internal stress, post-corrosion bond, RBSM, rib height

## Abstract

The buildup of corrosion products over a reinforcing bar and associated reduction in rib height lead to degradation of the bond between reinforcement and concrete. The authors have previously used digital image correlation (DIC) to visualize and quantify load-induced cracking at the interface in specimens with varying degrees of corrosion. The results obtained in that study are used here to simulate the post-corrosion local bond behavior. A bond degradation model is incorporated into the discrete analysis tool, 3D Rigid Body Spring Model (RBSM) for the simulation. This analysis method allows the shape of the reinforcing bar to be directly modeled, and concrete cracking behavior is simulated by using a randomly shaped mesh. The magnitude of opening and sliding over the tips of ribs in the simulation, in which the reduction in rib height could not be modeled, is significantly lower than observed in the experiment. The results demonstrate that reduction in rib height is an important factor in post-corrosion behavior, and needs to be included in simulation models. It is also understood that in order to gain a better understanding of local post-corrosion bond behavior, de-bonding between reinforcement and concrete needs to be modeled in a discrete analysis framework.

## 1. Introduction

The bond between reinforcement and concrete is a crucial factor governing the deformation and resistance of reinforced concrete structures against different types of loading. Various studies have been conducted to understand and model the effect of physical, structural, and material parameters, such as bond length, concrete cover to reinforcing bar diameter ratio, etc. on bond [1,2,3]. Overall, the bond interaction consists of two major components, namely mechanical interaction due to the shape of ribs and frictional interaction due to the surface roughness of the reinforcing bar. In the uncorroded case, bond failure occurs either by crushing of the concrete between ribs or by wedging action. In this case, the type of failure depends on rib geometry, i.e., the relative rib area, rib height, rib slope and rib spacing [4,5,6].

However, corrosion of the reinforcement over time is manifest as bond degradation. Corrosion reduces the reinforcement rib height, thereby reducing mechanical interaction, while the accumulation of corrosion products at the interface increases slipping. The combination of these behaviors is referred to as rebar damage [7,8,9]. The expansive corrosion products induce cracking in the concrete cover (referred to as concrete damage), which in turn causes loss in confinement around the reinforcing bar. Studies have shown that post-corrosion bond behavior is affected by the degree of corrosion, the absence or presence of stirrups, the ratio of cover to bar diameter, and experimental conditions, etc. [10,11,12,13]. The macro-scale and local damage induced in reinforced concrete by corrosion can lead to changes in the response of a structure. Corrosion inhibits the transfer of tensile stress from reinforcement to concrete, thus resulting in loss of tension stiffening and changes in deformation behavior [14,15,16]. Furthermore, in the case of long structural elements, reinforcing splicing is necessary, and bond plays a very important role in governing behavior. Corrosion of reinforcement in the splice region modifies cracking and deformation behavior, as well as load carrying capacity [17]. Furthermore, the degree of corrosion and its distribution has been observed to influence failure mode and the degradation of shear behavior in reinforced concrete beams [18,19,20].

Since bond behavior clearly has a significant impact on the structure, it is important to develop numerical simulation models that predict and help understand the post-corrosion structural response to varied loadings. In the past, various finite element models have been developed to simulate this post-corrosion response [8,21,22]. Furthermore, complex continuum models that simulate the entire corrosion deterioration mechanism, including the transport of deleterious ions, de-passivation of reinforcement and bond degradation, have also been developed [23,24]. However, in these continuum models, bond degradation is simulated by either reducing the tensile strength of the concrete or modifying the stress-strain curve for the post-cracking stage. However, studies have shown that loss in confinement due to cracking plays a significant role in governing post-corrosion behavior [11,25]. Therefore, there is a need for a numerical simulation model in which the effect of loss in confinement around a reinforcing bar due to concrete cracking and the effect of reduction in rib height can be directly modeled.

Apart from continuum models, there exist various discrete models for simulating the concrete fracture phenomena. Concrete is discretized, and its response to various kinds of loading has been successfully modeled in the past by Cusatis et al. [26], Schlangen and van Mier [27] and Aydin et al. [28]. Separately, a 3D Rigid Body Spring Model (RBSM) for reinforced concrete was developed by Nagai et al., based on the foundations laid by Kawai [29]. The model was further upgraded by Jiradilok et al. [30] to simulate-post corrosion bond behavior by incorporating a corrosion expansion model and bond degradation model into the 3D RBSM framework. The current framework has been successfully used at the macro level to simulate the response of corroded beams [31] and investigate the effect of stirrups on corrosion-induced cover cracking and the rate of bond degradation [13]. However, corrosion of reinforcement is non-uniform, and causes changes in deformation and stress conditions at the local level. To verify the performance of the simulation model at the local level at the interface, an experimental scheme to directly observe and visualize post-corrosion bond interactions using Digital Image Correlation (DIC) was developed by Avadh et al. [32].

DIC has been used to directly observe bond interaction at the interface for uncorroded specimens [33]. DIC is a non-contact image processing method where certain points are identified in the region of interest, and their deformation, simultaneously to the application of load, is tracked and measured. This allows very small deformations to be measured without the need for sensors such as strain gauges. The experimental scheme developed by Avadh et al. [32], based on the studies by Okeil et al. [33] and Jiradilok et al. [30], uses the DIC technique to directly record bond interaction between corroded reinforcement and concrete. As mentioned earlier, corrosion of reinforcing bar leads to damages in concrete and reinforcing bar. The combined effect of these two damages modifies the bond interaction in a complicated manner. Hence, in order to enhance the understanding develop and improve bond degradation models, it is important the separate the two damages. The effect of loss in confinement around the reinforcing bar due to corrosion induced concrete cracking has been investigated and simulated in detail [11,13], thus it is necessary to investigate the effect of damage in reinforcing bar. The objective of the previous experimental study was solely to investigate the effect of rib height reduction and corrosion product accumulation on local bond interaction. To achieve this, corrosion was first induced in the reinforcement using the impressed current technique. The corroded reinforcement was then cast into a concrete specimen with the geometry shown in Figure 1, where a window in the center allows direct observation of the interface during load application [32]. A notch was added in the center of the window to ensure cracking initiated at the center, and not from the window edges. Uniaxial tensile tests were performed on the specimens, while a video camera was placed directly in front of the window recorded interactions simultaneous with the application of loading. Using DIC, cracking at the interface was quantified through the opening and sliding relationship over the rib tips, and at the flat parts between the ribs. These quantitative measurements, which reflect the change in failure mode from mechanical interaction to sliding failure, are available for use to investigate whether the current 3D RBSM framework is able to offer improved simulations of post-corrosion local bond interaction.

## 2. Simulation Method

In this section, the details of the 3D RBSM framework utilized to simulate the post corrosion bond degradation has been described.

### 2.1. 3D Rigid Body Spring Model (RBSM)

In 3D RBSM, the concrete and steel are discretized into three-dimensional rigid bodies which are connected to adjacent bodies by one normal and two shear springs, as shown in Figure 2a. To diminish the effect of mesh bias in cracking, concrete is randomly meshed with a size in the range of 10 × 10 × 10 mm^3^ to 20 × 20 × 20 mm^3^, as shown in Figure 2b. Furthermore, due to the uniform mechanical behavior of steel, the reinforcement is discretized using regularly shaped rigid bodies, such that its actual shape, along with ribs, is incorporated into the model, as shown in Figure 2b,c.

### 2.2. Constitutive Model

The mechanical properties of concrete and reinforcement are governed by defining constitutive relationships for the normal and shear springs. The normal springs for concrete do not fail in compression and have a tri-linear tensile characteristic, as shown in Figure 3a. On the other hand, an elasto-plastic model is adopted for the shear springs, as shown in Figure 3b,c and calculated using Equation (2). The constitutive model for the normal springs connecting steel rigid bodies is shown in Figure 3d, with shear springs behaving elastically and governed by Equation (3). The normal and shear springs connecting concrete and steel rigid bodies at the interface are governed using the same relations as for concrete, but the tensile strength of these springs is half of that of springs connecting two concrete rigid bodies.
(1)$$\tau_{\max}=\pm (1.6f_{t}^{2}{\left(-\sigma +f_{t}\right)}^{0.4}+0.15f_{t})\;if\;(-3f_{t}\geq \sigma \geq f_{t})$$
(2)$$\tau_{\max}=\pm (1.6f_{t}^{2}{\left(4f_{t}\right)}^{0.4}+0.15f_{t})\;if\;(\sigma <-3f_{t})$$
(3)$$\sigma_{s}=E_{s}\varepsilon_{s}\;if\;\left(\varepsilon_{s}<\varepsilon_{y}\right)\;\sigma_{s}=f_{y}\;if\;\left(\varepsilon_{y}<\varepsilon_{s}<\varepsilon_{sh}\right)\;\sigma_{s}=f_{y}+\left(1-e^{\frac{\varepsilon_{sh}-\varepsilon_{s}}{k}}\left(1.01\;f_{u}-f_{y}\right)\right)\;if\;\left(\varepsilon_{s}>\varepsilon_{sh}\right)$$

### 2.3. Bond Degradation Model

The current 3D RBSM framework is able to simulate loss in confinement around the reinforcing bar due corrosion expansion induced concrete cracking [13,30,31]. However, since the objective of this study is to analyze the effect of rib height reduction and presence of corrosion layer at interface, only bond degradation model has been applied here. Reinforcement corrosion leads to a reduction in the cross-section of the reinforcement, which causes a reduction in yield load and reduction in rib height. Additionally, the increased surface roughness increases frictional interaction at the interface [30]. The effect of rebar damage on bond degradation was isolated in the experimental study by Jiradilok et al. [30], and based on their experimental results, two modification factors were proposed with respect to the degree of corrosion (x), namely F_normal_ and F_shear_, as shown in Figure 4. Since the geometry of the reinforcement cannot be changed mid-computation, corrosion induced degradation of the material and physical properties of the reinforcing bar is simulated by modifying the normal and shear springs using these two parameters.

F_normal_ is applied to normal springs connecting reinforcement and concrete rigid bodies at the interface, which reduces their tensile strength and stiffness, thereby inducing earlier fracture and simulating reduction of mechanical interaction. Furthermore, parameter F_normal_ is also applied to reduce the tensile strength of normal springs connecting reinforcement rigid bodies. Since the geometry of the reinforcement cannot be changed, F_normal_ reduces the yield strength, thus reducing the yield load and simulating reduction in cross-section. Frictional interaction between corroded reinforcement and undamaged concrete increased, as per the experimental observations by Jiradilok et al. [30], and thus the parameter F_shear_ increases the tensile strength of the shear springs at the interface.

In this study, the computation does not include a corrosion expansion model to simulate corrosion-induced concrete cracking, since in the experimental study being simulated, the corroded reinforcing bar was initially corroded and then re-casted into a new specimen, thus eliminating the effect of corrosion induced cracking on bond interaction.
(4)$$F_{\mathrm{normal}}\left(\%\right)=100-x$$
(5)$$F_{\mathrm{shear}}\left(\%\right)=200-\frac{100}{0.5x+1}$$
where *x* is the degree of corrosion (%) represented as a percentage reduction in mass after corrosion.

## 3. Simulation Scheme and Models

The current section provides background information regarding the details of the experimental study selected for simulation, geometry and boundary conditions of the simulation model, as well as the considered analysis cases.

### 3.1. Considered Experimental Study

The simulation model is based on the experiments conducted by Avadh et al. [32]. In the study, the reinforcing bar of 25 mm diameter was corroded by applying a current density of 150 μA/cm^2^ until the required degree of corrosion was achieved. Following this, the reinforcement was prepared and re-casted into a new specimen with a 100 mm × 100 mm cross-section and a 400 mm length, with geometry as shown in Figure 1. The new specimen had a window in the center to directly observe bond interaction while loading. Adding a window in the center of the specimens causes sudden changes in the stress distribution along the reinforcing bar due to a break in symmetry and a decrease in effective bonding area, thus modifying bond behavior. However, previous studies have shown a limited effect on the global response of the specimen. Hence, it is an acceptable arrangement, as it facilitates direct observation of bond interaction at the interface [32]. A uniaxial tensile load was applied to the new specimen by fixing on one side and pulling from another, as in the test setup shown in Figure 5.

### 3.2. Simulation Model

To allow simulated local bond responses to be compared with the experimental study, the exact geometry of the specimen as cast in the experimental study by Avadh et al. [32] is modeled. Thus, the model has dimensions of 400 mm × 150 mm × 150 mm, and there is an 80 mm-wide observation window in the center, as shown in Figure 6. The concrete is discretized using a mesh size of 10 × 10 × 10 mm^3^ to 20 × 20 × 20 mm^3^; however, a finer mesh size of 5 mm is used on the surface of the concrete within the window to enable accurate capture of the cracking phenomenon and crack pattern induced by bond interaction, as shown in section D-D. The elements in the center of the window are meshed into finer square-shaped elements, as the specimen had a notch in the center as shown in section C-C, to ensure cracking from the center. The diameter of the reinforcing bar is 25 mm. The material properties used in the simulation are identical to those of the experimental study, as shown in Table 1. The model has, in total, 13,200 elements consisting of 11,100 concrete elements and 2100 steel elements.

In the simulation, the steel bar is elongated at a rate of 0.001 mm for the first 100 steps and 0.005 mm for the remaining 980 steps, to cause a total elongation of 5 mm. The initial rate is kept slower, in order to obtain accurate internal stress conditions during the crack initiation phase.

### 3.3. Analysis Cases

The objective of this study is to investigate the effect of a reduction in rib height and the presence of a corrosion product at the interface using the current bond degradation model in 3D RBSM. Hence, to conduct this investigation and identify the improvements required in this bond degradation model, cases with an extreme degree of corrosion were selected. From the experimental study, two specimens are selected for modeling, as shown in Table 2: UC-00 (uncorroded specimen); and C-20 (specimen with corrosion of 20%). Two simulations are carried out for investigation and comparison, one with a degree of corrosion of 0% (SUC-00) and the other with 20% corrosion (SC-20). The post-corrosion bond behavior is simulated using the modification factors proposed by Jiradilok et al. [30], and described in Section 2.3.

## 4. Results and Discussion

This section discusses the findings from simulation of uncorroded and corroded cases, and compares them with the experimental observations.

### 4.1. Load-Slip Behavior

This section discusses the local pullout behavior observed in the simulation, and compares it with the observed experimental behavior. Figure 7 shows the specific points selected for calculation of slipping at the interface. These points and the equations used for calculation are identical to those used in the DIC experiment. Only the points on the left-hand side of the notch are selected, because in the experiment, pullout was observed on that side. Displacements are only measured along the X direction, which is the direction of load application. Slip is calculated by measuring the local displacement of the marked points, as per Equations (6)–(8) [32].
(6)$$Steel\;Elongation:{\;\Delta }_{s}=S_{2X}-S_{1X}$$
(7)$$Concrete\;Elongation:\;\Delta_{c}=C_{2X}-C_{1X}$$
(8)$$Slip:s=\Delta_{s}-\Delta_{c}$$
where, *S*_1*X*_, *S*_2*X*_, *C*_1*X*_, and *C*_2*X*_ are the displacements (in mm) in the X direction for the corresponding rigid bodies representing reinforcement and concrete, as shown in Figure 7.

The load-slip relationships for the two simulation models SUC-00 and SC-20 (solid lines) and the corresponding experimental specimens (dotted lines) are shown in Figure 8. In the simulation results, it can be observed that the load level at which the rapid increase in slip begins is lower when there is corrosion. However, the slope of this part of the load-slip response is the same with or without corrosion, despite the degraded stiffness of the reinforcing bar based on parameter F_normal_. Comparing the uncorroded simulation model SUC-00 and experimental specimen UC-00, the response is seen to be somewhat similar. In the corroded cases, the experimental results show that, with corrosion, the amount of slip with respect to load (i.e., the slope of the plot) is higher, but this phenomenon is not observed in the simulation. On the other hand, the magnitude of the load at failure is similar in the corroded model SUC-20 and experimental specimen C-20. That is, the bond degradation model in 3D RBSM is able to simulate the reduction in ultimate load due to corrosion, but not the local deformation behavior.

### 4.2. Crack Propagation and Strain Distribution Using DIC

Figure 9a,b shows the simulated cracking pattern and shear strain distribution for load levels of 20 kN, 40 kN, 60 kN and 80 kN at the window surface for uncorroded and corroded cases. Similarly, Figure 9c,d shows the normal strain distribution, with blue corresponding to tension and red to compression. Local deformations have been magnified by 10. The experimentally observed cracking patterns are shown for comparison in Figure 9e,f. At 20 kN, the higher strain along the center of the observation window, i.e., along the notch and perpendicular to the reinforcing bar, is observed, along with a strain in the diagonal direction emanating from the tip of each rib towards the notch. As the load increases to 40 kN and then to 60 kN, the diagonal strains further increase and diagonal cracks widen in both SUC-00 and SC-20. At 80 kN, the primary crack has completely split the corroded model SC-20 into two halves, with diagonal cracks on the surface. In the experimental specimens, it should be noted that specimen UC-00 had sharp ribs, while in corroded specimen C-20 the ribs had disappeared due to corrosion, as shown by the rib profile in Figure 10a,c, respectively. The diagonal cracking pattern due to mechanical interaction observed in specimen UC-00 (Figure 9e(iv)) is well simulated by model SUC-00 (Figure 9a(iv),c(iv). However, due to the absence of ribs, corroded specimen C-20 does not show any diagonal cracks, as shown in Figure 9f(i–iv)), in contrast with the uncorroded specimen UC-00. Hence, failure in the corroded case is primarily due to slipping. The simulation model does not allow for the shape of the ribs to be altered according to the degree of corrosion. Thus, F_normal_ and F_shear_ are used to simulate post-corrosion response. Due to the presence of ribs in corroded model SC-20, diagonal cracks are induced owing to the ongoing mechanical interaction, so its behavior differs from that in the experiments.

The diagonal cracks observed in models SUC-00, SC-20 and specimen UC-00 represent mechanical interaction between reinforcement ribs and surrounding concrete. Figure 10 shows detailed views of the failure at the interface between reinforcement and concrete, as observed in the experiment and modeled in the simulations. In uncorroded specimen UC-00, diagonal cracks can be seen over the tips of the ribs, and clear de-bonding between the reinforcing bar surface and concrete can be observed. In the uncorroded model, SUC-00, diagonal cracks similar to those seen experimentally are observed, but the surrounding concrete rigid bodies remain attached to the steel rigid bodies. Hence, the local bond failure mechanism in the simulation differs from reality. Corroded specimen C-20 failed primarily due to slip, as shown in Figure 10c, and there was no diagonal cracking even with the presence of the very small ribs. Again, clear de-bonding can be seen between the reinforcing bar and surrounding concrete. However, the failure cracking pattern for corroded model SC-20 shows that even though bond capacity is reduced and failure occurs earlier (as shown in Figure 8), the concrete elements remain mostly attached to the steel elements. Thus, very little de-bonding occurs.

The stiffer response of the corroded model, as discussed in the previous section, can be attributed to the fact that, in the simulation, the de-bonding between concrete and steel elements is different from that observed in the experimental study.

### 4.3. Local Opening and Sliding Behavior

In this section, the simulation results are compared with the local opening and sliding behavior, as measured in the experiment. That is, the change in local response due to implementation of modification factors F_normal_ and F_shear_ is investigated. In the simulation, the opening and sliding behavior is measured by selecting steel and neighboring concrete rigid bodies, as indicated by points B1 and A1, respectively, in the upper section of Figure 11.

The displacement of these points in the X and Y directions was extracted, and opening and sliding were then calculated using Equations (9) and (10) in a similar way to that used in the experimental study. Only the rib tips in the top left segment were selected for comparison.
(9)$$Opening=|B_{iY}-A_{iY}|$$
(10)$$Sliding=|B_{iX}-A_{iX}|$$
where, *A_iX_* and *A_iY_* are the displacements of point *A_i_* in the X and Y directions, respectively, in mm, *B_iX_* and *B_iY_* are displacements of point *Bi* in the X and Y directions, respectively, in mm, as extracted from the 3D RBSM simulation framework.

Figure 11 compares the opening and sliding observed in the experiment and that calculated from the simulation. For the experimental results, the average of the opening and sliding measurements is plotted as a red line in Figure 11a for the uncorroded case (UC-00), and as a brown line in Figure 11b for the corroded case (C-20). Comparing UC-00 and C-20, it can be observed that, with corrosion, the magnitude of opening decreases and sliding increases, owing to the reduced rib height and the accumulation of corrosion products.

The opening and sliding relationship for uncorroded model SUC-00 and corroded model SC-20 is shown in Figure 11a,b, respectively. From the plot, it is very difficult to determine a clear trend. It should also be noted that the uncorroded model shows greater sliding than opening. The maximum value of opening in the uncorroded model SUC-00 is only around 0.12 mm, as compared to 0.6 mm observed in the experiment. Hence, in simulation, local opening above the tips is less than that observed in the experiment. Similarly, in corroded model SC-20, the maximum value of opening and sliding is 0.04 mm and 0.4 mm, respectively, which is very small compared to the corroded specimen C-20. Overall, the simulation models predict very much less opening and sliding than experimentally observed. The low magnitude of opening in the simulation may be attributed to the fact that the simulation framework is unable to efficiently simulate de-bonding between reinforcing bar and concrete. In the considered experimental study, the corrosion of reinforcing bar causes reduction in rib height, and forms a layer of corrosion product over the corroded region. Therefore, on re-casting the specimen, despite elimination of corrosion-induced concrete cracking, de-bonding was easier at the interface. Meanwhile, in the simulation, the rib height could not be changed in the discrete model. Therefore, significant amount of mechanical interaction continues to occur, thus making de-bonding difficult and restricting the opening and sliding. This is in coherence with the discussion in the previous section. Hence, it can be concluded that accurate simulation of post-corrosion bond behavior requires improvement in modelling of the de-bonding behavior at the interface.

## 5. Conclusions

This study highlights the need to model corrosion-induced reduction in reinforcement rib height and the debonding process at the interface to better understand the post-corrosion bond behavior. The 3D RBSM framework was used to develop a model with the same geometry as a previous experimental study, with parameters F_normal_ and F_shear_ used to simulate post-corrosion bond behavior. From the simulations carried out in this study, the following conclusions can be derived:Load-slip results from the simulation are similar to those measured in the experiment for the uncorroded case. However, for the corroded model, the reduction in load at failure is appropriately simulated, but the simulated response was much stiffer than that seen in the experiment.Through DIC-derived strain maps, it was determined from the experiment on a corroded specimen that mechanical interaction had been degraded as a result of the corrosion-induced reduction in rib height. In this case, no diagonal cracking was manifest. However, in the simulated corroded model, since rib height cannot be reduced mid-computation, mechanical interaction continued to take place.Detailed observation of failure at the interface showed that, in the experiment, bond failure occurred due to clear de-bonding between reinforcement and surrounding concrete. On the other hand, in the simulation, no such de-bonding was observed even in the corroded model, as the concrete elements remained attached to their neighboring steel elements.In comparing the opening and sliding relationships obtained from the simulation at the rib tip to the experiment results, it was observed that the simulated maximum value of opening is significantly less than that in the experiment, even for the uncorroded model. It is clear that as concrete elements remain attached to steel elements in the simulation, opening above the tip of the ribs is restricted.

These results clarify that to enhance understanding of post-corrosion local bond behavior, it is important to model the corrosion-induced reduction in rib height. However, with regards to the prediction of average post-corrosion bond capacity, the current 3D RBSM model performs satisfactorily.

## Figures and Tables

**Figure 1 materials-15-06470-f001:**
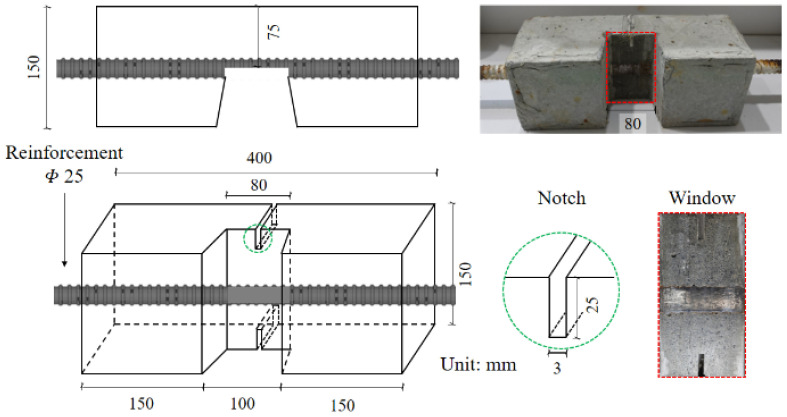
Geometry of re-cast specimen with details of notch and window Reprinted with permission from [32]. 2021, Elsevier.

**Figure 2 materials-15-06470-f002:**
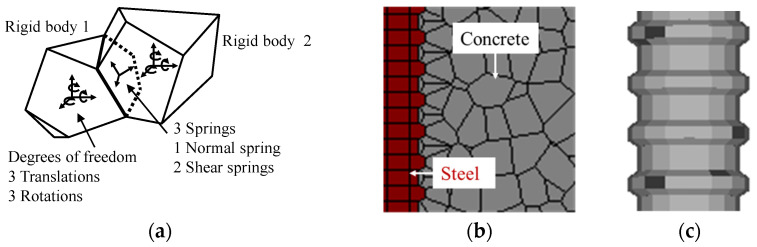
Discretization of the reinforced concrete model in 3D RBSM Reprinted with permission from [13] 2021, Elsevier. (**a**) Polyhedral rigid bodies interconnected by springs; (**b**) cross-section of steel and concrete elements; and (**c**) 3D rebar model.

**Figure 3 materials-15-06470-f003:**
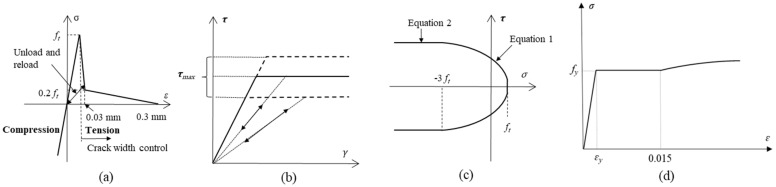
Constitutive laws governing the behavior of (**a**) normal springs in concrete (**b**) elasto-plastic behavior for shear spring in concrete (**c**) calculation of maximum shear stress for shear springs for concrete and (**d**) tensile behavior of reinforcing bar in 3D RBSM Reprinted with permission from [13]. 2021, Elsevier.

**Figure 4 materials-15-06470-f004:**
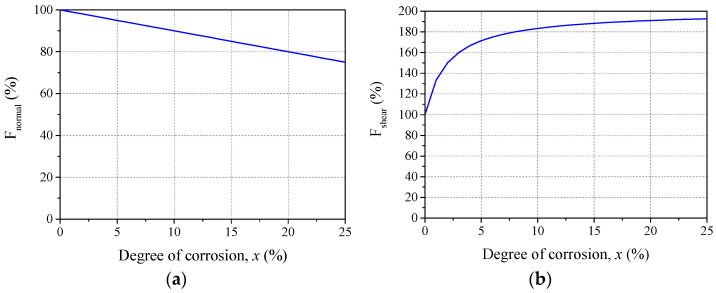
Variation of parameters according to degree of corrosion: (**a**) Modification factor for normal springs; and (**b**) modification factor for shear springs.

**Figure 5 materials-15-06470-f005:**
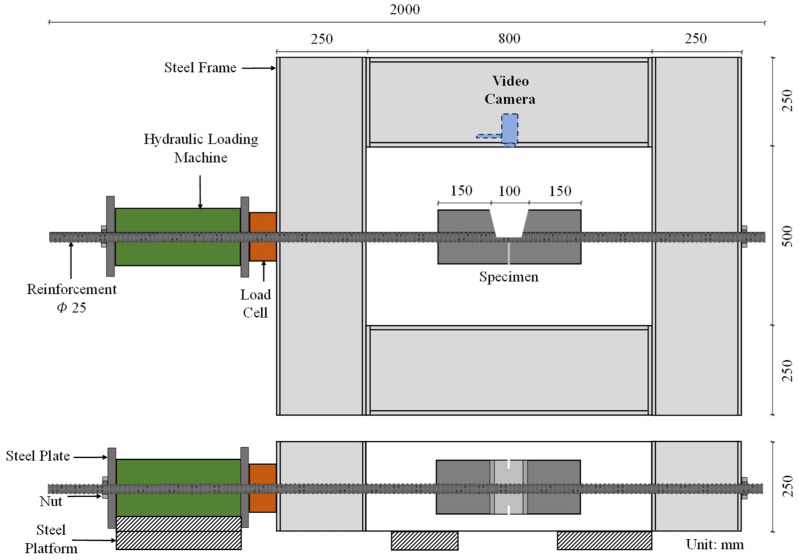
Experimental setup to conduct uniaxial loading test on the new specimen. Reprinted with permission from [32]. Copyright year 2021, Elsevier.

**Figure 6 materials-15-06470-f006:**
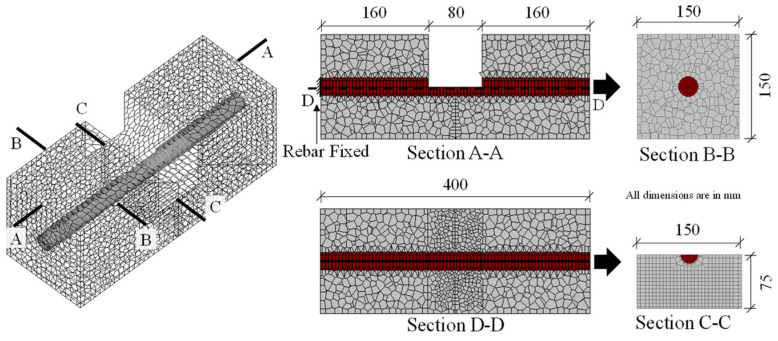
Geometry of model with window and notch in 3D RBSM.

**Figure 7 materials-15-06470-f007:**
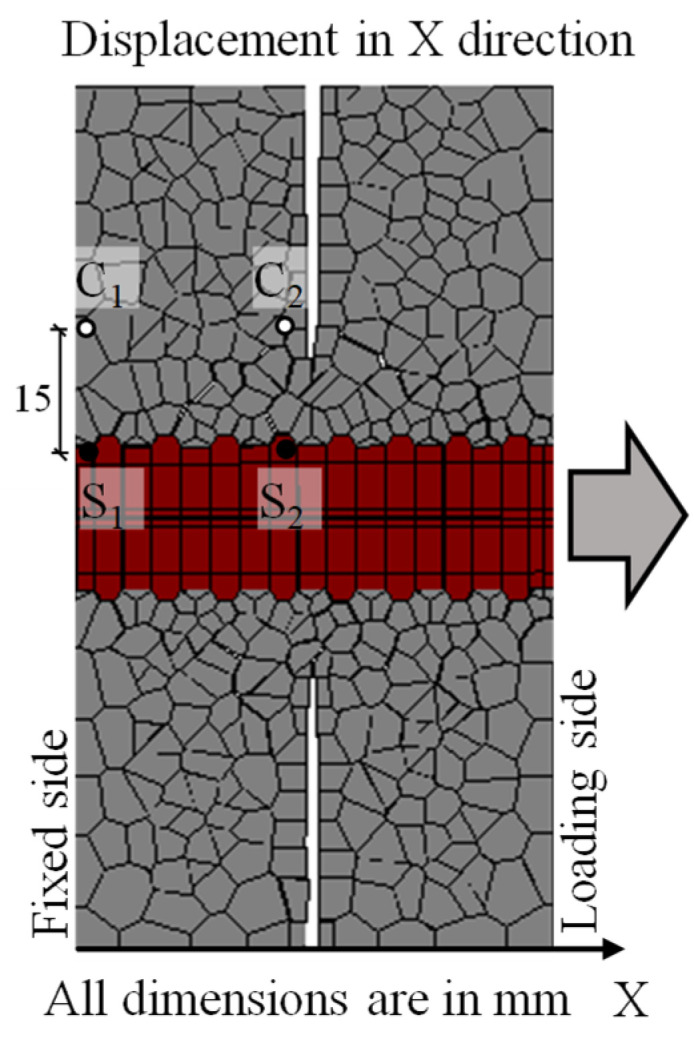
Location of points in concrete and steel at interface for calculation of slip in simulation.

**Figure 8 materials-15-06470-f008:**
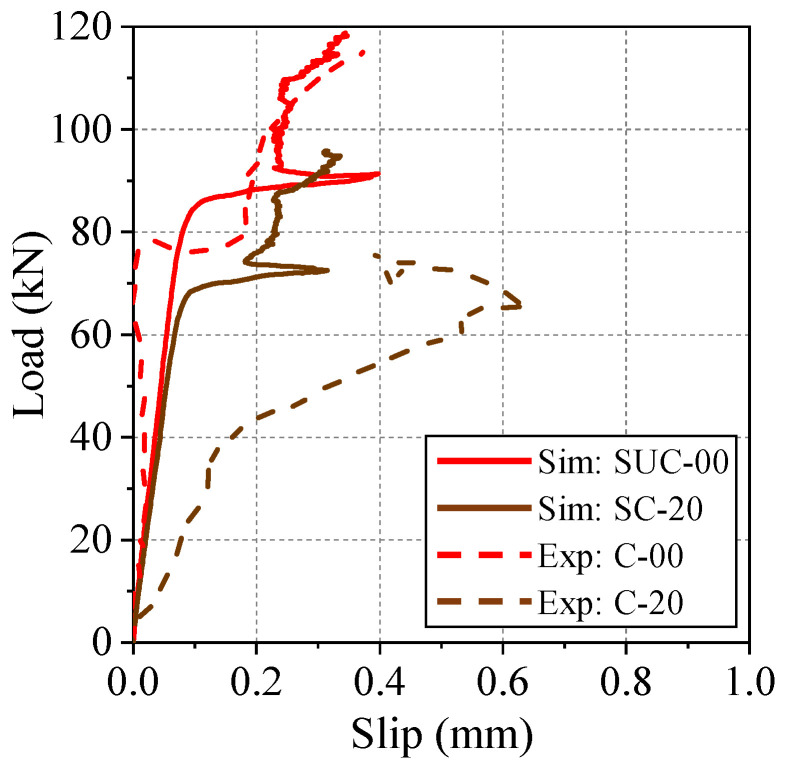
Comparison of slipping behavior between simulation and experiment.

**Figure 9 materials-15-06470-f009:**
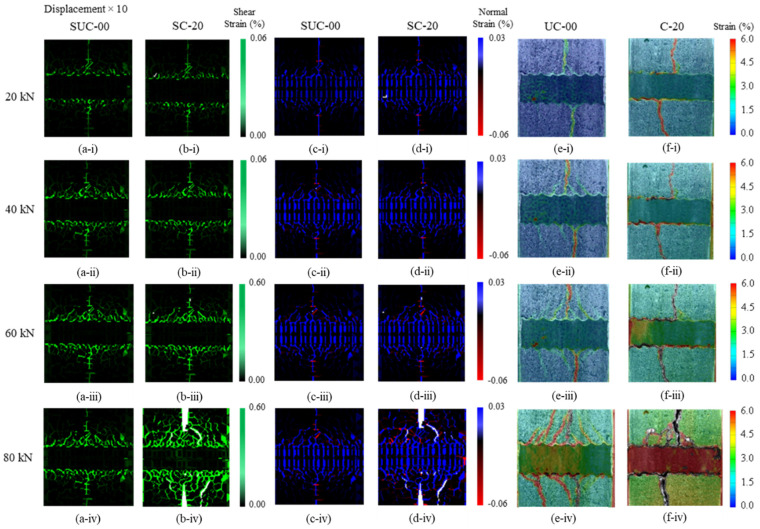
Change of strain distribution with loading in uncorroded and corroded cases (**a**,**b**) shear strain in simulation, (**c**,**d**) normal strain in simulation, (**e**,**f**) observed using digital image correlation (DIC) in experiment.

**Figure 10 materials-15-06470-f010:**
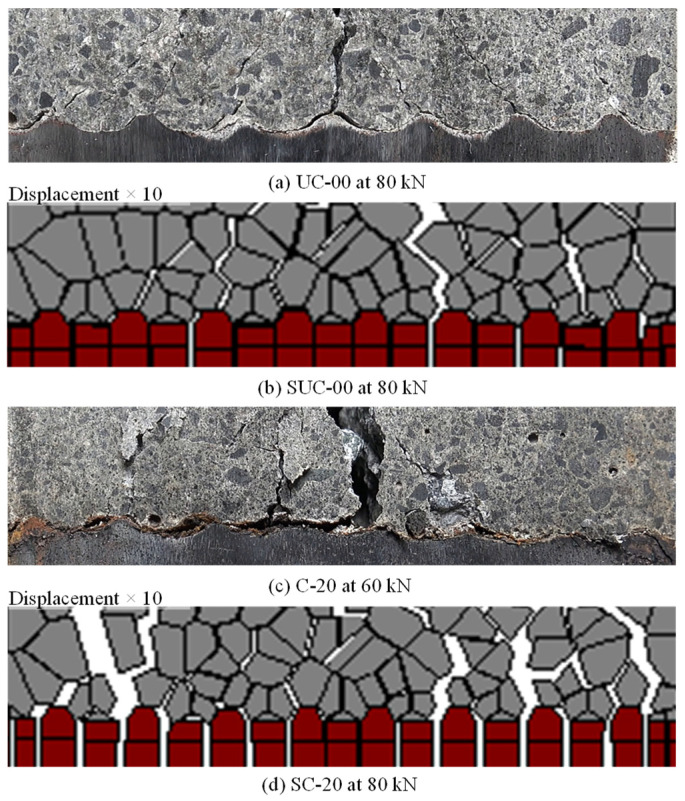
Observed final cracking pattern in: (**a**) uncorroded case in experiment; (**b**) uncorroded case in simulation; (**c**) corroded case in experiment; and (**d**) corroded case in simulation.

**Figure 11 materials-15-06470-f011:**
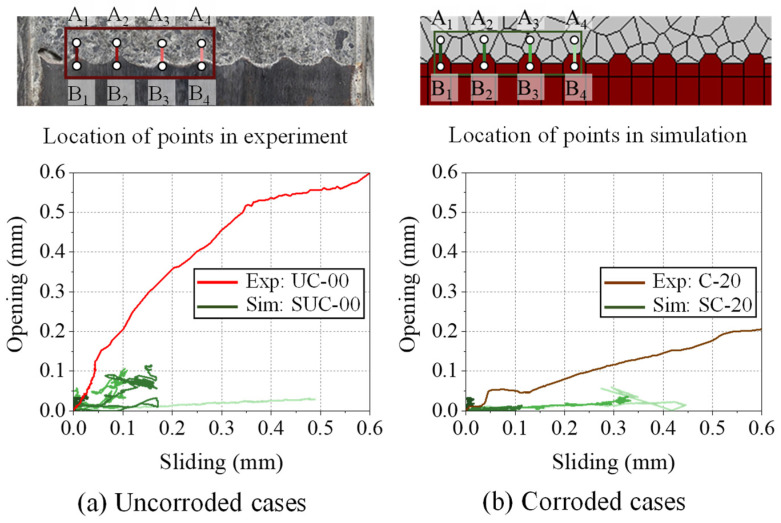
Location of points and opening-sliding relationship observed in the experimental simulation study for.

**Table 1 materials-15-06470-t001:** Input material properties.

**Property**	**Concrete**	**Reinforcing Bar**
Tensile strength (MPa)	2.6	-
Elastic modulus (MPa)	27,000	190,000
Yield strength (MPa)	-	345

**Table 2 materials-15-06470-t002:** Analysis cases.

Experiment		Simulation	
Specimen	Degree of Corrosion (x%)	Model	Degree of Corrosion (x%)
UC-00	0	SUC-00	0
C-20	20	SC-20	20

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
