# Peer review of "3D RBSM Analysis of Bond Degradation in Corroded Reinforced Concrete as Observed Using Digital Image Correlation"

_materials, 2022, doi:10.3390/ma15186470_

Round 1

Reviewer 1 Report

 In this paper, the 3D RBSM framework was used to simulate the corrosion induced debonding process in reinforced concrete, and the simulated results were compared with experimental observation via digital image correlation. The topic and content are interesting. However, the manuscript need to be revised as follows:

1. The figure numbers should be correct.

2. Expand figure captions so figures are almost self-explanatory.

3. Writing needs improvement. Each section should open with a paragraph or two explaining what to expect before going to subsections.

4. Although this paper focus on modelling work, some necessary description of experimental procedure also need to be added.

Reviewer 2 Report

The authors have presented a very well-written article about an area of interest to the practicing engineer, especially as global infrastructure continues to age. The authors do present the strengths and limitations of their design, though I feel there are a few areas where the authors need to be clearer in this regard.

One item I would like to see addressed by the authors concerns the specimen design. The window cut into the concrete will provide a portion of the bar where corrosion products will not induce tensile stress in the concrete, potentially altering the bond behavior. This is an unavoidable effect of needing to use DIC, but I'd like to see its effects discussed.

Corroding the reinforcement prior to casting in concrete will remove the effect of tensile stress from the buildup of corrosion products, as the authors note in the introduction. Why was this method chosen as opposed to corroding the bar cast in concrete?

The concrete mesh size seems large relative to the spacing of ribs on most reinforcement. Did the authors conduct a sensitivity analysis of the effect of mesh size?

20% corrosion is quite high, and well above what will cause significant damage in reinforced concrete. Did the authors consider intermediate cases?

Reviewer 3 Report

Dear Authors,
thank you for your paper focused on the analysis of bond degradation in corroded RC using digital image correlation.
My comments are:
- Figure 1 is followed by Figure 1 again, it should be Figure 2 - renumber all Figures,
- line 26, you use "response ... again different types of loading" - what do you mean by "response"? Respons to load are load effect (internal forces as moments MEd, shear forces VEd, normal forces NEd, etc.), but structure/member has resistance (ultimate bending resistance moment MRd, etc.), in this case, it is not better to use "resistance"?
- lines 92-95, line 103, line 159 etc. - it is better to use "hole" (in member) as "window",
- fig. 1 and fig. 6 (wrong denotation fig. 5) - missing information on how the samples (experimental and numerical simulations) were loaded. Were they subjected to normal tensile force? Were the reinforcements pulled at both ends, or was it a pull-out test? Or were they loaded perpendicular to the sample - bending occurred (3-point or 4-point bending test)?
- the text lacks more information about the experiment, was it presented in another article?
- line 106, fig. 2 (second Fig. 1 in paper), there is used "rigid body" - I think that it is better to use "rigid solid",
- line 132, line 144 - Jiradilok is [30] or [31], not [15],
literature [15] and Russo et al.,
- line 146-148 - it is stated that this study neglects the effect of corrosion expansion, which is probably a mistake because the literature states that it is possible up to a level of about 5%. In this study, however, corrosion up to 20% is considered, so it should probably not be neglected.
- Figure 10 (wrong denotation figure 9) - do the gaps between the reinforcement elements in the numerical model mean that the reinforcement has broken?
- Figure 11 (wrong denotation figure 10) - there are considerable differences between experimental measurements and numerical simulations (especially in the case of (a) uncorroded cases - how do you explain this? Is the numerical model correct? It is not due to the fact that corrosion was not considered an expansion model?

Best regards.

Round 2

Reviewer 1 Report

My comments have been addressed.

Reviewer 3 Report

Dear Authors,

thank you for improving your paper. I have no other comments.

Best regards.